# Use of Tenofovir Alafenamide/Emtricitabine/Elvitegravir-Cobicistat in HIV-Naive Patients with Advanced Disease: GENIS Study

**DOI:** 10.3390/jcm11174994

**Published:** 2022-08-25

**Authors:** Javier Perez Stachowski, David Rial Crestelo, Ana Moreno Zamora, Noemi Cabello, Pablo Ryan, Nuria Espinosa Aguilera, Otilia Bisbal, Maria Jesus Vivancos Gallego, Maria Jose Nuñez, Jesus Troya, Montserrat Dominguez, Julian Olalla Sierra

**Affiliations:** 1Internal Medicine Department, Hospital Costa del Sol, 29603 Marbella, Spain; 2HIV Unit, Hospital Universitario 12 de Octubre, 28041 Madrid, Spain; 3Department of Infectious Diseases, Hospital Ramón y Cajal, 28034 Madrid, Spain; 4Department of Infectious Diseases, Hospital Clínico San Carlos, 28040 Madrid, Spain; 5Internal Medicine Department, Hospital Infanta Leonor, 28031 Madrid, Spain; 6Department of Infectious Diseases, Hospital Virgen del Rocío, 41013 Sevilla, Spain

**Keywords:** severe immunosuppression, antiretroviral treatment, HIV

## Abstract

Objective: The primary endpoint of the study was to determine the proportion of patients with HIV RNA < 50 copies/mL at 48 weeks. Design: Phase IV, multicentric, open-label, single-arm clinical trial of participants recruited in 2018–2019 to evaluate the efficacy and safety of tenofovir alafenamide/emtricitabine/elvitegravir-cobicistat (TAF/FTC/EVG-c) as first-line treatment in HIV-1 infected naïve participants with advanced disease. Methods: Adverse events were graded according to the Division of AIDS scale version 2.0. Quantitative variables were recorded as median and interquartile range, and qualitative variables as absolute number and percentage. T-Student or Wilcoxon tests were used to analyze intragroup differences of the continuous variables. Results: Fifty participants were recruited with a baseline median CD4 lymphocyte count of 116 cells/µL and a viral load of 218,938 copies/mL. The proportion of patients with viral load <50 copies/mL at week 48 was 94% in the per-protocol analysis, with a median time of 1.9 months to achieve it. Three adverse events attributed to the study drug caused trial discontinuation. Conclusions: the use of TAF/FTC/EVG-c in patients with advanced HIV disease in our study demonstrated efficacy comparable to data from pivotal clinical trials with a good safety profile.

## 1. Introduction

Advanced HIV disease is defined as the presence of <200 CD4 lymphocytes/µL or an AIDS-defining condition before the start of antiretroviral treatment (ART) [1]. The presentation with such severe immunosuppression is clearly associated with an increase in morbidity and mortality, especially in the first year of diagnosis [2,3,4], leading to higher healthcare costs and a greater possibility of viral transmission [5,6]. The proportion of naïve patients presenting with severe immunosuppression is still very high; in Spain, 28.1% had ≤200 CD4 lymphocytes/µL at diagnosis in 2018 [7]. There is little evidence regarding this type of patient [8,9,10], and, furthermore, there are no specific recommendations in the antiretroviral treatment guidelines beyond the specific precautions on drug interactions in the case of concurrent mycobacterial infection [11].

The number of patients with severe immunosuppression included in clinical trials is low, especially in trials of naive patients with the most modern drugs. Traditionally, these patients were preferred candidates to receive protease inhibitors because of their high genetic barrier since treatment was started before receiving the resistance test. However, drugs such as raltegravir, with a lower genetic barrier than protease inhibitors, showed a high degree of virologic efficacy in patients with severe immunosuppression [12]. The combination of TAF/FTC/EVG-c in a single tablet showed a virologic suppression in around 86% of 112 patients in its pivotal studies [13].

## 2. Materials and Methods

### 2.1. Design

Phase IV, low-intervention, open-label, single-arm clinical trial designed to evaluate the efficacy and safety of TAF/FTC/EVG-c as first-line treatment in severely immunosuppressed HIV-1 infected naïve patients. Recruitment began in April 2018 and ended in May 2019, with the last study visit being recorded in June 2020. The trial was approved by the Ethics Committees of the participating centers and the Spanish Medicines Agency and was carried out in accordance with the Declaration of Helsinki and good clinical practice guidelines. The candidates were informed of the study objectives and procedures, so they signed informed consent before their inclusion. The trial was registered at Clinical Trials.gov (NCT03693508).

Patients were recruited from six Spanish centers. Eligible patients needed to be at least 18 years old with positive serology (Western Blot) for HIV-1, naïve to antiretroviral treatment, and <200 CD4 lymphocytes/µL. They had to present serum transaminase levels ≤5 times the upper limit of normality, >1000 neutrophils/µL, >50,000 platelets/µL, >85 g/L of hemoglobin, and serum amylase levels <1.5 times the upper limit of normality. No candidate was excluded because of an active opportunistic infection.

After signing the informed consent and extraction of screening samples, the scheduled visits were carried out at baseline and at weeks 4, 8, 12, 24, 36, and 48. At the baseline visit, treatment with TAF (10 mg)/FTC (250 mg)/EVG-c (150–150 mg) in a single daily tablet was initiated. At each visit, clinical data, vital signs, blood count, lipid profile, liver enzymes, kidney function, CD4 lymphocytes, and HIV-1 viral load were collected in the local laboratory.

Clinical and laboratory adverse events were evaluated by local investigators and graded according to the AIDS division scale (version 2.0) [14]. The protocol defined virologic failure as the presence of two consecutive viral loads >50 copies/mL with at least two weeks of separation if the patient had previously reached undetectability; or two consecutive viral loads >1000 copies/mL from week 24. In the case of genotypic resistance, a test was performed. The primary endpoint of the study was to determine the proportion of patients with HIV RNA < 50 copies/mL at week 48. Secondary endpoints included the increase in CD4 lymphocytes, the proportion of participants with virologic suppression, and the immune recovery rate, defined as median until achieving CD4 lymphocytes > 200/µL. Safety endpoints included the incidence of grade 3–4 clinical or laboratory adverse events and treatment interruptions due to adverse events.

### 2.2. Statistical Analysis

In the intention-to-treat (ITT) analysis, all those patients who received at least one dose of the drug were considered for the analysis, this being the reference population for the safety analysis. In the per-protocol population analysis (PPP), we considered patients who had received at least one dose of the drug without resistance mutations to any of them and did not show major deviations from the protocol, such as treatment dropout, adherence < 80%, or discontinuation of antiretroviral treatment due to prohibited interactions. Quantitative variables are shown as the median and interquartile range (IQR), and qualitative variables as absolute number and percentage. The 95% confidence intervals of the results associated with the primary objective and the main secondary variables are presented. The intragroup differences of the continuous variables were analyzed using the t-Student test or Wilcoxon for paired samples, depending on the normality of the variables. The time to virologic failure or suppression and the cumulative probabilities of these events have been estimated using the Kaplan–Meier method. The statistical package SPSS 26.0 (IBM Corp., Armonk, NY, USA) was used for the analysis of the data, and the value of statistical significance was established at *p* < 0.05 (two-tailed).

## 3. Results

Between April 2018 and June 2019, 50 patients were recruited, 84% males and 90% Caucasians with a median age of 37 years. There was an AIDS event at diagnosis in 12 (24%) participants, *P. jiroveci* being the most frequent (6 participants). The baseline CD4 lymphocytes were 116/µL (59–157), and the viral load was 218,938 copies/mL (70,976–652,900).

The reasons for trial discontinuation are presented in Table 1. Only two patients with good adherence had a viral load >50 copies/mL at week 48, without the development of resistance mutations in genotypic tests. The proportion of patients with less than 50 copies/mL at week 48 was 62% for ITT and 94% for PPP. In PPP, the median time to achieve viral load <50 copies/mL was 1.9 months.

There was a progressive increase in the number of CD4 lymphocytes; thus, the median CD4 at weeks 4 and 48 were, respectively, 201 and 375 cells/µL. The median CD4 lymphocytes increase at week 48 was 220.5/µL (IQR: 143–294). At week 48, 88.1% of the patients had CD4 lymphocytes > 200/µL, and the median time to achieve it was 1.8 months.

The median weight of the patients changed from 68 kg at the baseline visit to 74 kg at week 48, and the body mass index (BMI) changed from 23.8 kg/m^2^ to 25.1 kg/m^2^ (*p* < 0.0005 in both cases). There was a slight decrease in eGFR (*p* < 0.01). At week 48, a statistically significant rise was observed in all lipid fractions, except for triglycerides, but with little change in the total cholesterol/HDL ratio. A total of 260 adverse events were recorded in 46 patients (92.0% of patients with a ratio of 5.7 events per patient). The most frequent was genital herpes in 3.6%. Regarding the relationship with the study medication, five adverse events occurred in five patients. Treatment had to be interrupted due to adverse events in five patients, three of them related to the study drug. No deaths were recorded.

## 4. Discussion

Our results show the efficacy of the combination of TAF/FTC/EVG-c similar to that reported in pivotal clinical trials in patients with <200 CD4 lymphocytes/µL [13]. These results are of special relevance since patients with severe immunosuppression are underrepresented in clinical trials, and, nevertheless, they are the ones with the highest mortality and morbidity after starting antiretroviral treatment [15]. The short period of time to achieve undetectability and CD4 lymphocyte count >200/µL (less than two months in both cases) should be noted, leading patients to a clear decrease in the possibility of opportunistic infections in the first six months after starting antiretroviral therapy [16]. The percentage of undetectability at week 48 is higher than the ones reported with protease inhibitors and efavirenz [8,10] and similar to those reported with other integrase inhibitors [9]. A large recently published real-life cohort study yielded similar data on virologic efficacy at twelve months in patients “on treatment” with the use of integrase inhibitors regardless of the degree of immunosuppression [17], although in this case, the limit to define viral suppression was set at 200 copies/mL.

Despite the number of reported AIDS events, no deaths occurred in our study. Although the change in BMI in our patients was not very marked, going from normal weight to slightly overweight, it is true that there were two cases of treatment interruption. Weight gain with current treatment regimens is a subject of discussion, especially in severely immunosuppressed patients where “return to health” could play a greater role. Otherwise, the combination of TAF/FTC/EVG-c was well tolerated with few treatment interruptions. There were no renal, hepatic, or bone side effects that led to the discontinuation of treatment. Regarding the lipid profile, although there was an increase in LDL cholesterol and triglycerides, the total cholesterol/HDL ratio decreased slightly, which would argue in favor of the fact that there is no increased cardiovascular risk with the introduction of TAF/FTC/EVG-c.

TAF/FTC/EVG-c was a nationally and internationally recommended first-line treatment at the time this study was designed. Despite the difficulty of using an enhancer such as cobicistat and not having a genetic barrier as high as second-generation integrase inhibitors, the results of TAF/FTC/EVG-c in this population were successful, with few viral failures and without resistance mutations developed.

There are several limitations to our study. Firstly, it is a single-arm study, so we have to refer to indirect comparisons in terms of efficacy. Secondly, although it provides greater experience in treating this type of patient, the sample size is limited. The difficulty in recruiting and keeping these patients in clinical trials is that some are completed before reaching the projected sample size [9], and others redefine their objectives [8].

## 5. Conclusions

The use of TAF/FTC/EVG-c in our study showed comparable results to the data extracted from its pivotal clinical trial in patients with severe immunosuppression with a good safety profile.

## Figures and Tables

**Table 1 jcm-11-04994-t001:** Reasons for discontinuation before week 48.

Reasons of Discontinuation	*n*
Abandonment	6
Only baseline visit completed	2
Per protocol withdrawal	5
Basal mutations	1
Prohibited treatments	1
Study investigator’s decision	1
Consent withdrawal	1
Pregnancy	1
Secondary events	2
Wight gain	1
Dizziness	1
Treatment adherence < 80%	4

## Data Availability

The data presented in this study are available on request from the corresponding author. The data are not publicly available due to privacy restrictions.

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
