# Peer review of "Use of Tenofovir Alafenamide/Emtricitabine/Elvitegravir-Cobicistat in HIV-Naive Patients with Advanced Disease: GENIS Study"

_jcm, 2022, doi:10.3390/jcm11174994_

Round 1

Reviewer 1 Report

Study is clear, data are clearly presented, text also ok, conclusions are correct.

Only fig. 1 needs improvement, as for example Y-axis is not labelled.

Reviewer 2 Report

The paper summarizes the results of a Phase IV clinical trial (NCT03693508). The trial was performed between Apr 2018 to May 2019. The results are interesting to readers.

However, the English of the paper causes misunderstanding.

For example:

Line 68 “<1.5 times the limit higher normal”

Line 68-70 “No candidate was excluded for presenting active opportunistic infection, unless its treatment contraindicated the use of TAF / FTC / EVG / cb.”

Line 98 “did not show resistance mutations to any of the drugs used,”

Line 112-114 “There was an AIDS event at diagnosis in 12 (24%) participants, being P.jiroveci the most frequent (6). In 45 (90%) the selection and baseline visits were performed on the same day.”

Line 128-129 “The proportion of undetectable individuals in the different visits is reflected in Figure 1.”

What is the population of for ITT, ITTm and PPP? Line 126-127 , the results state as “ The proportion of patients with less than 50 copies / mL at week 48 was 62% for ITT, 79% for ITTm, and 94% for PPP. ”

line 196-197 mentioned: "Supplementary Materials: The following supporting information can be downloaded at: www.mdpi.com/xxx/s1, Figure S1: title; Table S1: title; Video S1: title. " however, the Supplementary Materials are not included in the draft. 

Minor comments:

Please keep consistency: such as: CD4+ or CD4 , microL or µL; TAF/FTC/EVG-c or TAF / FTC / EVG / cb

Typo:

Line 38: ” ” should be  µL

Reviewer 3 Report

This study by Stachowski JP Et.al was to evaluate the efficacy and safety of tenofovir alafenamide /emtricitabine / elvitegravir-cobicistat (TAF/FTC/EVG-c) as first-line treatment in HIV-1 infected naïve participants with advanced disease.

There are some issues

Please present the result section line no 111-129 in a tabular format and showing sample size, eligibility, percentage of dropout etc. The presentation of results section could have been lot better. Please improve it.

There are few type errors. Please consider correcting it. For examples

In line no 33 “microL” please write either microliters or abbreviation

Line no 38 there is a type error etc.

Round 2

Reviewer 3 Report

The Authors have addressed my concern. 

However, there are still lots of typo error. I would advised to check the manuscript thoroughly.